# The Importance of Cytokinins during Reproductive Development in Arabidopsis and Beyond

**DOI:** 10.3390/ijms21218161

**Published:** 2020-10-31

**Authors:** Giada Callizaya Terceros, Francesca Resentini, Mara Cucinotta, Silvia Manrique, Lucia Colombo, Marta A. Mendes

**Affiliations:** Dipartimento di Bioscienze, Università degli Studi di Milano, 20133 Milano, Italy; giada.callizaya@unimi.it (G.C.T.); francesca.resentini@unimi.it (F.R.); mara.cucinotta@unimi.it (M.C.); manrique.silvia@unimi.it (S.M.); lucia.colombo@unimi.it (L.C.)

**Keywords:** cytokinins, reproduction, seed formation

## Abstract

Fertilization and seed formation are fundamental events in the life cycle of flowering plants. The seed is a functional unit whose main purpose is to propagate the plant. The first step in seed development is the formation of male and female gametophytes and subsequent steps culminate in successful fertilization. The detailed study of this process is highly relevant because it directly impacts human needs, such as protecting biodiversity and ensuring sustainable agriculture to feed the increasing world population. Cytokinins comprise a class of phytohormones that play many important roles during plant growth and development and in recent years, the role of this class of phytohormones during reproduction has become clear. Here, we review the role of cytokinins during ovule, pollen and seed formation at the genetic and molecular levels. The expansion of knowledge concerning the molecular mechanisms that control plant reproduction is extremely important to optimise seed production.

## 1. Introduction

Cytokinins (CKs) play a crucial role in regulating many aspects of plant growth and development [1,2,3]. Endogenous CKs are adenine derivatives and are categorised into two groups, based on whether they possess an isoprenoid or aromatic side chain. Isoprenoid CKs are more widespread in nature and include isopentenyladenine (iP), trans-zeatin (tZ)-, cis-zeatin (cZ)- and dihydrozeatin-type derivatives. Generally, tZ and iP exhibit higher activities than cZ, and in addition to their sugar conjugates, tZ and iP are the most abundant forms, although much variation exists, depending on the plant species, tissue and developmental stage [4,5].

The initial and rate-limiting step in CKs biosynthesis is catalysed by isopentenyltransferases (IPTs). *Arabidopsis thaliana* possesses nine *IPT* genes: *AtIPT1*, *AtIPT3* and *AtIPT4–AtIPT8* encode ATP/ADP IPTs, whereas *AtIPT2* and *AtIPT9* encode tRNA isopentenyltransferases, which modifies a subset of adenine bases on tRNAs [6]. The subsequent biosynthesis step involves the cytochrome P450 enzymes CYP735A1 and CYP735A2, which convert CK nucleotides into the corresponding tZ-nucleotides [7,8]. Finally, enzymes encoded by the *LONELY GUY* (*LOG*) gene family activate CKs by converting nucleotides into the corresponding nucleobases [9]. CKs are degraded by CYTOKININ OXIDASE/DEHYDROGENASES (CKXs) and the Arabidopsis *CKX* gene family contains seven members (*AtCKX1* to *AtCKX7*), each of which shows distinct spatio-temporal patterns of expression and slightly different enzymatic properties [10,11]. In addition to degradation, CKs can also be deactivated by glucosylation by UDP-glucosyl transferases (UGTs) [12,13]: N-glucosylation is irreversible, whereas O-glucosylation can be reversed by β-glucosidases (GLUs). 

CKs are perceived via a multistep phosphorylation pathway. In Arabidopsis, the three histidine protein kinases (AHKs) AHK2, AHK3 and AHK4/CRE1 function as CK receptors. Upon CK binding, AHK2-4 are auto-phosphorylated and a phosphoryl group is transferred to histidine phosphotransfer proteins AHPs (in Arabidopsis, AHP1–5). AHPs can also be phosphorylated by CYTOKININ INDEPENDENT 1 (CKI1) that acts independently of CKs because it lacks the cytokinin-binding domain [14]. In turn, AHPs transmit the phosphorelay signal to nuclear response regulators ARRs, which modulate the transcription of cytokinin-related genes. Two classes of ARRs exist: nuclear type-B response regulators that act as positive transcriptional regulators, and type-A regulators, which are involved in negative feedback loops [15,16] that also inhibit the transcription of type-B ARRs. In addition, a large proportion of the CK transcriptional response is also promoted by cytokinin response factors (CRFs), a group of closely related transcription factors belonging to the AP2 family [17].

Although CK signalling and perception pathways have been characterised in great detail (Figure 1) and increasingly more studies in recent years have highlighted the importance of CKs during many developmental and physiological processes, how CKs control these developmental processes at the molecular level is still not fully understood. In this review, we discuss the most recent insights regarding the role of CKs during reproductive development and mainly focus on ovule, pollen and seed development in the model species *Arabidopsis thaliana.*

### 1.1. Cytokinins Play a Pivotal Role in Ovule Patterning and Development

Flowering plants alternate between a diploid sporophytic stage, which constitutes the main body of the plant, and a reduced, haploid gametophytic stage contained within the female and male floral organs. The female gametophyte (FG), or embryo sac, develops within the maternal sporophytic tissues of the ovule, which provides structural support to the female gametes and encloses them until the seed develops, following fertilization [18]. 

Ovule primordia arise from the meristematic placenta tissue within the gynoecium, and mutants in which the synthesis or perception of CKs is compromised, produce fewer ovule primordia [19]. By contrast, increased levels of CKs in *ckx3ckx5* double mutant lead to an increase in gynoecium size and to the formation of more ovules, due to increased meristematic activity of the placenta tissue [20,21]. Moreover, it has been also reported for other Brassicaceae species that CKs influence ovule number, suggesting that the genetic manipulation of CK metabolism is a potentially powerful tool to enhance seed yield [19,22]. Intriguingly, however, it was recently demonstrated that high levels of CKs in *ckx7* mutants inhibited fruit elongation [23].

In flowering plants, megasporogenesis begins when ovules arise from the placenta as finger-like structures. Within the ovule primordia, a single subdermal nucellar cell enlarges and differentiates into a megaspore mother cell (MMC) (Figure 2a). The MMC then undergoes meiosis to generate four haploid megaspores, three of which degenerate and the remaining one becomes the functional megaspore (FM) (Figure 2a). During this process, the two inner and outer integuments surrounding the developing female germline develop and after three rounds of mitosis, the FM gives rise to the FG in a process called megagametogenesis (Figure 2a).

CKs are important from the very beginning of ovule emergence and several studies have demonstrated that disruption of CK homeostasis affects all phases of ovule development. For instance, integument initiation in the triple CK receptor mutant *ahk2-2 ahk3-3 cre1-12/ahk4* is impaired and some ovule primordia remain as long finger-like structures, with most of the remaining ovules arresting during gametogenesis [24,25,26]. Specification of the FM is also affected in this triple mutant, which does not express the *pFM1::GUS* identity marker, and some ovules display a FG that contains only one or two nuclei, or only a central cell-like nucleus, which explains the observed block in gametogenesis [25]. Promoter studies revealed that during ovule development, *AHK3* and *CRE1/AHK4* are preferentially expressed in the chalaza region, whereas *AHK2* is expressed throughout the ovule (Figure 2c) [24,25]. In conclusion, disruption of the three-cytokinin histidine kinase receptors (AHKs) clearly affects ovule integument and FG development, although the effect varies, depending on the *ahk* mutant alleles, and the phenotypes of all mutant combinations are summarised in Table 1. Defects in ovule integument development have also been observed in wild type ovules exogenously treated with CKs (6-*Benzylaminopurine*, BAP), which developed a single structure instead of two integuments. Notably, only the *cre1-12*/*ahk4* mutant developed two integuments when treated with BAP, whereas integument development remained affected in the other single receptor mutants *ahk2* and *ahk3*, demonstrating a major role for CRE1/AHK4 in the detection of excessive CK levels [24]. The genetic network that controls ovule development involves other two key transcription factors, *BELL1 (BEL1)* and *SPOROCYTELESS/NOZZLE* (*SPL/NZZ*), and the *bel1 spl* double mutant possesses finger-like ovule structures without integuments, similar to those in *ahk2-2 ahk3-3 cre1-12* [24]. The expression of both *BEL1* and *SPL* is regulated by CKs, although CKs negatively affect *BEL1* expression and positively affect that of *SPL*. In addition to a role in integument formation, the fundamental role of *SPL* in MMC initiation has been well characterised and ovules of *spl* fail to produce male and female germlines [27]. Moreover, *SPL* regulates the expression of *WUSCHEL*, which encodes a CK-responsive homeodomain transcription factor initially identified as a stem-cell regulator in the shoot meristem. *WUS* is also required for megasporogenesis and its loss of function leads to defective MMC development [28]. Several studies have demonstrated that CKs control the expression of the auxin efflux carrier gene *PIN-FORMED 1* (*PIN1*), which is required to create the auxin gradient necessary for ovule primordium formation and integument growth. In particular, *PIN1* expression in the placenta and ovule primordia is mainly mediated by CRFs, because it is reduced in the *crf2 crf3 crf6* triple transcription factor mutant affecting CK signalling [29]. In the chalaza, auxin–cytokinin crosstalk converges on the regulation of *PIN1* by SPL and BEL1. In *spl* mutant ovules, *PIN1* is not expressed, whereas it is ectopically expressed in *bel1*. In *pin1-5*, a few ovule primordia develop as finger-like structures, similar to the ovule phenotype of double *spl bel1* and triple *cre1-12 ahk2-2 ahk3-3* mutants [24].

### 1.2. CK Perception Influences Female Gametophyte Cell Identity

In wild type Arabidopsis ovules, the FG derives from three rounds of mitosis. The first two rounds occur without cytokinesis and lead to a four-nucleate coenocyte FG with two nuclei at each pole. During a third mitotic division, phragmoplasts and cell plates form between sister and non-sister nuclei; this represents the beginning of the cellularisation process and the FG cells quickly become completely surrounded by cell walls. All these events lead to the formation of a coenocytic, eight-nucleated embryo sac. Subsequent nuclear migration, polar nuclear fusion and cellularisation take place to ultimately produce a seven-celled FG that consists at the micropylar pole of two synergids, one egg cell, one diploid central cell, and three antipodal cells at the chalazal pole [30,31,32].

The histidine protein kinase CKI1 involved in CK perception is required to specify cell identity in the FG. Experiments with Arabidopsis protoplasts have shown that *CKI1* overexpression leads to the cytokinin-independent activation of the two-component CK signalling pathway [33] and to constitutive activation of the *type-A RESPONSE REGULATOR 6* (*ARR6)* promoter [34]. In addition, CKI1 shares downstream components with the cytokinin signalling pathway, such as the His phosphotransfer proteins AHP2 and AHP3 that act upstream of ARRs in the CK signalling cascade [35]. The mutant alleles *cki1-5* and *cki1-6* cause lethality [36]. As suggested by the genetic heritability of the *cki1-5* and *cki1-6* alleles, half of the FGs in heterozygote plants were defective [36]. Deng and colleagues in 2010 partially rescued the lethal female gametophyte phenotype of *cki1-5* using the *CKI1* promoter to drive the expression of either *IPT8* (biosynthetic gene) or *ARR1* (a type-B Arabidopsis Response Regulator- CK signaling/response), demonstrating that CKI1 is required for cytokinin signalling and for appropriate FG development [35]. 

More recently, in 2016, a fourth mutant allele, *cki1-9*, was described, whose embryo-sac formation was also compromised [37]. The use of specific FG identity markers showed that both the central cell and the accessory antipodal cells in *cki1-9* lose their identity in favour of egg-cell identity, synergid cell identity was correctly maintained. In *cki1-9* mutants, the ectopic egg cell can be fertilized, the seeds contained zygotes without developing endosperm, consistent with loss of central cell identity [37]. By contrast, ectopic expression of CKI1 under control of the FG-specific (*pAKV*) promoter leads to mis-expression of the central cell marker that became visible in the egg and synergid cells position. The mutant ovules developed seeds with dual endosperm but no embryos [37]. A similar phenotype was observed when *CKI1* was expressed specifically by egg cell- or the synergid cell-specific promoters. The *CKI1-GFP* protein fusion is expressed from the beginning of FG formation and throughout its development. At the one- and two-nucleus stages, CKI1 co-localises with the FG-specific endoplasmic reticulum (ER) marker, *pAKV::TaqRFPer*, in the perinuclear regions. When the nuclei at each pole divide during the second mitotic event to form the four-nucleate FG, CKI1 localisation becomes polarised and is only detected in the ER surrounding the two sister nuclei at the chalazal pole, distinguishing between the nuclei of the chalaza and micropyle [37]. In the mature FG, CKI1 expression is restricted to the central and antipodal cells (Figure 3).

As mentioned above, CKI1 shares downstream components with the CK signalling pathway: BiFC protein–protein interaction and yeast two-hybrid assays demonstrated that CKI1 interacts with the CK signaling AHP2, AHP3 and AHP5 and more weakly with AHP1 proteins [38,39]. The *AHP* genes are expressed in the embryo sac (Figure 2c); in particular promoter analysis using nuclear eGFP demonstrated that *pAHP1* is active in the central cell and the synergid cells (Figure 3), *AHP2* and *AHP5* are both expressed in all female gametophytic cells, whereas *AHP3* is specifically expressed in the central cell [40] (Figure 3). Only *AHP2*, *AHP3* and *AHP5* appear to be involved in FG formation. The *ahp2 ahp3 ahp5/+, ahp2 ahp3/+ ahp5, and ahp2/+ ahp3 ahp5* multiple mutants reproduced *cki1-9* mutants, with a strongly reduced sensitivity to cytokinin and reduced fertility and produced gametophytes in which antipodal and central cells identity was lost, and egg-cell fate was acquired [40]. Furthermore, the quintuple mutant siliques were shorter, and seed development was variable, because seed abortion was often observed. However, the mutant seeds produced were larger than those in wild type [41]. The two primary classes of type-A and type-B response regulators (ARRs) respond to CK signalling. Both ARR types can be phosphorylated, but only type-B ARRs contain a Myb-like DNA binding domain [42]. The type-A ARRs are rapidly induced by CKs and compete with type-B ARRs for phosphorylation, but function as negative regulators of the cascade [34,43] (Figure 1). The quadruple type-B ARR mutant *arr1-3 arr2-2 arr10-2 arr12-1* produces a substantial number of ovules with arrested gametophytes [25], similar to ovules in multiple *ahk* and *cki1* mutants. Type-A ARR double mutant *arr7arr15* also cause female gametophytic lethality, although the cause of this lethality was not investigated [44].

The specification of FG-cell identity also requires two members of the REPRODUCTIVE MERISTEM (REM) family of transcription factors, VERDANDI (VDD) and VALKYRIE (VAL). *VDD* and *VAL* are expressed throughout all stages of ovule development, and their mutation causes a strong defect in synergid-cell specification (Figure 3). The *vdd-1* homozygous mutant is lethal and heterozygous siliques contain a high percentage of unfertilized ovules. A similar phenotype was observed for *VAL-RNAi* transgenic lines. In *vdd-1* heterozygous and *VAL-RNAi* mutants, the identity of the synergids is partially lost, and a low percentage of the synergid cells in *vdd-1/+* ovules express the antipodal cell marker [45,46]. Synergid cells express *CKX7,* whereas the CKX7 protein is also detected in the egg cell [47], which is important for CK degradation, but the activity of *pCKX7::nlsGUS* was absent in approximately half of the *vdd1* heterozygous and *VAL_RNAi* ovules, suggesting that gametophytes that carry these genes knockdown, do not express *CKX7* and probably contain higher levels of active CK. This has been suggested to cause the persistence of the synergids after pollen tube arrival and thereby to affect sperm delivery and fertilization.

Another well-known mutant that exhibits a defect in female gametophyte cell specification is *altered meristem program 1* (*amp1*). *AMP1* encodes a glutamate carboxypeptidase that is important for shoot apical meristem development and phytohormone homeostasis [48] and *amp1-1* and *amp1-2* T-DNA mutant alleles present high levels of CKs [49]. Loss of *AMP1* function in *amp1-10* and *amp1*-13 leads to supernumerary egg cells at the expense of the synergids, enabling the formation of twin embryos [50]. AMP1 is not expressed only in the sporophytic tissue (Figure 3), but also in the synergids and more weakly in the egg cell [50] (Figure 3). Morphological analyses of *amp1/+* heterozygous plants identified twin embryos and supernumerary egg cells only very rarely. This indicates that sporophytic *AMP1* expression is sufficient to prevent the cell-fate change of synergids and suggests that AMP1 might move between cells or be required for the production of a mobile signal necessary for synergid identity [50]. The presence of high levels of CKs and synergid cells that are abnormally specified in *amp1* mutants, is consistent with previous observations for *cki1*, *ahp* and *vdd* and *val* mutants. All the hypothetical interrelationships are depicted in Figure 3.

### 1.3. Cytokinins Are Involved in Sporophyte–Gametophyte Communication

CKs execute important functions at all stages in ovule development, and particularly during gametophyte cell-fate acquisition. However, not all the components of the CK machinery are expressed in gametophytic cells, but some are also present in sporophytic cells, suggesting that the acquisition of identity within the gametophyte requires CKs-based communication between the gametophyte and sporophyte.

One of the most useful tools available with which to understand CK signalling is the *TCS::GFP* marker developed by Muller and collaborators in 2008, which reflects the transcriptional activity of ARR type-B response regulators to CKs and utilises their DNA-binding motif cloned upstream from the GFP reporter gene [51]. In 2013, Zürcher and colleagues established an improved version of the *TCS* synthetic reporter using an extended version of the ARR type-B DNA-binding motifs that is more sensitive to CK in most tissues analysed [52]. The use of this TCS reporter has enabled CK signalling/response during the different phases of ovule development to be monitored (Figure 2d). The analysis of *TCSn* revealed that at the very beginning of ovule development, *GFP* expression is detected only in the basal part of the ovule that corresponds to the chalaza region, immediately below where the megaspore mother cell MMC differentiates. At the FM stage, the GFP signal is once more restricted to the sporophyte in the chalazal region of the ovule. The same pattern of expression is observed during mitosis and gametogenesis. At the mature ovule stage, when the FG is completely formed, GFP expression becomes detectable in the micropilar and chalazal poles (Figure 2d) [52]. However, the biosynthetic genes expression pattern suggests that CKs synthesis does not overlap completely with CKs response (Figure 2). In Arabidopsis, the only *AtIPT* gene that is expressed during ovule development is *AtIPT1*, which is expressed in the MMC, companion cells and the surrounding sporophytic nucellar tissues (Figure 2b). During FM stage, following meiosis, *pIPT1* is active within the haploid megaspore, in the surrounding sporophytic cells and funiculus. In the subsequent stages, *pIPT1* expression is detected in the developing FG [24] toward the micropylar pole (Figure 2b).

Among LOG biosynthetic genes at FG stage, *LOG1* is expressed in the transmitting tract within the pistil, whereas *LOG3, LOG4* and *LOG5* are expressed within the ovule. In particular, *pLOG3* is expressed in the funiculus, *LOG4* is present at FG stage in the sporophytic tissues at the chalazal pole (Figure 2b and Figure 3), whereas *pLOG5::GUS* expression is present in the mature ovule (Figure 2b) and *LOG7* was mainly observed in the counterpart of the FG, the pollen [53]. CK transporters play an important role in CK distribution in ovules. In Arabidopsis, the AtPUP proteins are involved in the import of CK from the apoplast to the cytoplasm, and responses to CK signalling during Arabidopsis development are constrained by the transporter PURINE PERMEASE 14 (PUP14), which is expressed in early ovule primordia at the finger-like stage [54]. The expression of *PUP14* is inversely correlated with that of *TCSn* and the loss of *PUP14* leads to ectopic cytokinin signalling. The PUP14 protein localises to the plasma membrane and acts by importing bioactive CKs, thus depleting apoplastic CK pools and inhibiting CK perception at the plasma membrane [54]. AMP1, the glutamate carboxypeptidase mentioned before, might also be an important factor in the CK-based communication between the sporophyte and the gametophyte. In mature ovules, AMP1 is expressed mostly in the integuments, as well as in the synergids [50]. Heterozygous *amp1/+* plants do not show defects in synergid identity, indicating that sporophytic *AMP1* expression is sufficient to confer synergid identity. Intriguingly, the cell-specific expression of AMP1 in the synergid, egg or even in the central cells can complement *amp1* synergid defects [50], suggesting that synergid specification requires an AMP1-dependent mobile signal that has to reach the synergids, but can be provided by any of the surrounding cells [50]. *amp1* mutants have higher levels of CK synthesis, and contain a 4- to 6-fold increase in Zt and iP CK, respectively, compared to the wild type [49], and the authors suggested that *IPT* genes might be negatively regulated by AMP1. However, Kong and collaborators did not observe differences in *TCSn::GFP* expression in *amp1* ovules [50], suggesting that some components of the CK signalling cascade might be downregulated in *amp1* ovules and prevent an increase in *TCSn::GFP* signal, even in the presence of higher levels of CKs. However, it cannot be excluded that AMP1 might contribute to synergid specification via a mechanism unrelated to CKs, as it has been previously described for the SAM, were AMP1 regulates the stem-cell niche in a CK-independent manner [55] (Huang et al., 2015). In conclusion, the asymmetric and non-overlapping distribution of many CK biosynthesis and signalling components in the ovule and mutants, suggests that CKs are involved in the sporophyte/gametophyte cross-talk required to form a ovule [25,52,56].

### 1.4. CK Signalling Influences Stamen Development, Anther Dehiscence and Pollen Viability

Male gametophyte development takes place within stamens, which are composed of a filament and an anther. Inside the anthers, the non-reproductive cells differentiate in specialized tissue layers, including the tapetum, that surround the sporogenous cells [57,58]. Here, two distinct and successive developmental phases, microsporogenesis and microgametogenesis, take place and lead to the production of the mature pollen grain (Figure 4). During microsporogenesis, the sporogenous cells, also called pollen mother cell, enter meiosis to generate tetrads of haploid microspores. This stage is completed when the callose wall surrounding tetrads degenerates and individual microspores are released [58,59]. The free microspores then go through two rounds of mitotic divisions. During pollen mitosis I, the microspore divides asymmetrically to produce a large vegetative and a small generative cell. The control of asymmetric cell division in the first pollen mitosis is essential for the correct cellular patterning of the male gametophyte because each of the resulting two daughter cells possesses unique gene expression profiles that confer their characteristic structures and cell fates [60]. Then, a second mitosis produces two twin sperm cells enabling double fertilization to produce the embryo and endosperm (Figure 4).

Few studies have analysed the role of CKs during pollen development and function. Arabidopsis and tobacco stamens treated with exogenous CKs produce very short filaments and the anthers do not reach maturity uniformly. Moreover, the small amount of pollen produced is unable to reach the stigma due to the short length of the filaments, causing sterility. If excess CKs inhibit stamen development at the same time as a lack of CK perception, anther dehiscence and pollen viability is impaired [61,62]. The anthers of *ahk2 ahk4* and *ahk3 ahk4* CKs receptor double mutants contain only a small amount of viable pollen grains and do not dehiscence [63]. Similarly, the stamen filaments of the *ahk2 ahk3 cre1/ahk4* triple mutant at anthesis elongated normally, but the anthers were smaller than those in wild type and do not dehisce [26]. The *ahk2 ahk3 cre1/ahk4* triple mutant produced only two- or three-lobed anthers instead of the typical four-lobed structure and the degeneration of the tapetum and the break between the septum and stomium were incomplete. In addition, the vascular tissues of the anther failed to form normally, and the number of pollen grains was reduced. The pollen of the triple mutant was morphologically normal but did not germinate in vitro as efficiently as wild type pollen and could not germinate on wild type stigmas [26]. This inability to germinate might be caused by incomplete maturation, due to incomplete tapetum degeneration [26].

The signalling CK-related gene *AHP4* is predominantly expressed in young flowers and plays an important role in anther formation. Four AHPs (AHP1, AHP2, AHP3 and AHP5) are functionally redundant positive regulators of CK signalling, whereas AHP4 may weakly negatively participate in some CK responses [41]. Plants that overexpressed *AHP4* showed reduced fertility, due to a lack of secondary cell wall thickening in the anther endothecium. Conversely, the anther walls of *ahp4* were more lignified than those in wild type, indicating that AHP4 negatively regulates thickening of the secondary cell wall of the anther endothecium, and provides insight into the role of CK secondary cell wall formation [64].

Concerning CK degradation, analysis of *pCKX5::GUS* expression showed that *CKX5,* which encodes an apoplastic CKX, is expressed in young developing stamen primordia, and later becomes confined to the central part of developing anthers. Before and during pollination, *CKX5* expression is restricted to the mature pollen grains [11]. An indication of the potential involvement of CKXs in pollen development derives from the analysis of plants that overexpress *CKX1* and *CKX3*, which show reduced pollen production, particularly in early flowers [11]. Very little additional information is available regarding CK-related genes that are specifically expressed in pollen. For example the CK transporter *AtPUP3* expression is restricted to pollen, implicating a potential role for AtPUP3 in the transport of purine derivatives during pollen germination and tube elongation [65], as observed for Petunia pollen [66]. Another pollen-specific gene is the response regulator *ARR2*, which promotes the expression of mitochondrial respiratory chain complex I (nCI), which is upregulated in pollen during spermatogenesis [67]. These data reveal that CKs are essential for stamen development, anther dehiscence and pollen development, but the downstream CK gene network involved in these developmental processes largely remains unidentified. Future studies should include further understanding the crosstalk between CKs and other hormones, such as jasmonic acid and brassinosteroids, and the interactions among hormone signals and environmental cues.

### 1.5. Cytokinins and Seed Size

Successful fertilization and seed formation require the delivery of two non-motile sperm cells into the female gametophyte by the pollen tube. The first step towards fertilization occurs when the pollen tube comes into contact with the micropyle and the receptive synergid degenerates concomitantly with pollen tube burst, allowing sperm cell nuclei delivery. Subsequently, one sperm nucleus will fuse with the egg cell to form the zygote, and the other will fuse with the central cell to generate the endosperm, forming the seed.

CKs influence also seed development. In addition to the role of CKs in regulating ovule number, which affects seed number, CK levels also affect seed size. The constitutive overexpression of several *LOGs*, CKs biosynthetic genes, including *LOG2, LOG4, LOG5, LOG7* and *LOG8*, leads to the production of larger seeds than those in wild type. When *p35S::LOG4* pistils were crossed with wild type pollen, they produced seeds as large as those resulting from self-pollination, whereas seeds produced from the wild type pistils pollinated with *35Spro:LOG4* pollen were phenotypically wild type, suggesting that CK activity in the female sporophyte controls seed size [53]. However, the effects of CKs on seed size have been mostly observed following the mutation of CKX genes, which catalyse the irreversible degradation of CKs, and several of the seven CKX family members in Arabidopsis have been linked with seed yield. *CKX3* and *CKX5* are involved in the differentiation of reproductive meristems and *ckx3 ckx5* double mutants show delayed meristem activity termination, leading to the formation of more and larger flowers harbouring more ovules that develop into seeds [20]. An apparently contradictory finding regarding seed size control is that low levels of CKs lead to larger seeds, whereas high levels lead to smaller seeds. Thus, the overexpression of *CKX1*, which reduces CK levels [11], leads to larger seeds. *CKX2* is a target of the *HAIKU* (*IKU*) genetic pathway, a leucine rich repeat (LRR) kinase involved in regulating endosperm size. A reduction in the level of CKX2 in *iku2* mutants leads to an increased CK concentration in the endosperm and a concomitant reduction in seed size [68]. Consistent with this finding, CK-insensitive mutants such as the triple receptor mutant *cre1 ahk2 ahk3* [69], the quintuple phosphotransfer protein mutant *ahp1 ahp2 ahp3 ahp4 ahp5* [41], and the triple positive response regulator mutants *arr1 arr10 arr12* [70], also produce larger seeds because these genotypes mimic the absence of CK. This effect of CKX enzymes on seed size has drawn attention as a mechanism to increase crop yields [71,72]; however, the molecular mechanisms downstream of CKs that lead to this phenotype are currently unclear.

### 1.6. Cytokinins during Reproduction in Crop Species

The engineering of crop plants to obtain greater yields has been a major focus of plant biologists and breeders, with the aim of ensuring food availability for an increasing world population [73]. To increase seed yield in crops, some of the most important traits include ovule number and development, pollen viability, fertility rate and seed size. Research on the model plant Arabidopsis has led to an unprecedented wealth of knowledge on plant development that can guide improvements in crop performance [74].

*Brassica napus* is a crop plant belonging to the *Brassicaceae* family that is widely cultivated globally due to the edible oil that is extracted from its seeds. It has been recently demonstrated that Arabidopsis and *B. napus* share well-conserved response mechanisms to CK treatment. Exogenous CK application in *B. napus* causes a reduction in stamen length and anthers that do not mature uniformly, such that few produce pollen. However, the pollen was not able to reach the stigma due to the short length of the filament, causing sterility [22]. On the contrary, during maize (*Zea mays*) reproductive development, male-sterile plants can also be obtained following the ectopic accumulation of CKX. Maize plants that ectopically express *CKX1* under the control of a maize pollen- or anther-specific promoter, exhibit rudimentary terminal structures at the apical meristem that lack recognisable male florets or spikelets, and hence a tassel [75]. The exogenous application of kinetin, a synthetic CK that is not degraded by CKX1, and thidiazuron (TDZ), a CKX inhibitor, partially restored male development in transgenic maize plants, demonstrating that CKs execute a crucial role in maize reproduction [75] (Table 2).

In rice (*Oriza sativa*), several mutants that affect CKs have been described; an example is the CK biosynthesis gene *LONELY GUY (LOG*), which is required to maintain meristem activity and its loss of function causes premature shoot meristem termination, resulting in a severely reduced inflorescence size, an abnormal branching pattern, and fewer floral organs [9]. The *Oslog-3* rice mutant produces flowers that contain six stamens and a slender pistil that lacks an ovule, or no pistil [76] and *Oslog-1* and *Oslog-4* single mutant flowers display a weak phenotype, with only one stamen and no pistil [77] (Table 2). A CRIPR/Cas9 gene-editing approach has been taken to characterise the role of the type-B response regulators (RRs) in CK signalling in rice [78] (Table 2). Mutant phenotypes associated with a decreased activity of rice type-B RRs and CK responses include defects in inflorescence architecture, flower development and fertilization. Triple *rr21/22/23* mutants displayed reduced fertility because of defective stigma development. Anthers from the triple mutant were developmentally normal and produced pollen with a high viability, but the carpels lacked the papillae on the stigma that aid pollen capture, hydration and guidance of the pollen tube towards the ovary, which was probably the basis for the poor grain filling. Moreover, single *rr24* mutants are infertile because of defective anther development [78]. Collectively, these results demonstrate that CK biosynthesis and response are extremely important for the correct development of female and male floral structures. Guo et al. 2018 [79] recently investigated the association between grain number and grain size in rice and showed that *GRAIN SIZE AND NUMBER* (*GSN1*), a protein kinase phosphatase, controls the trade-off between grain size and number. The loss-of-function *gsn1* mutant possesses larger grains than wild type, but less-branched inflorescences, leading to the production of fewer grains than wild type. *CKX2* is upregulated in *gsn1* and consequently, CK levels are reduced during early panicle development, which supports a role for *CKX2*/CKs in controlling grain size and panicle architecture in rice [80]. Similarly, variants of *TaCKX6-D1*, a wheat (*Triticum aestivum*) orthologue of *OsCKX2*, were significantly associated with grain weight but not grain number [72,81] (Table 2). Increasing levels of CK are linked to larger seeds in Arabidopsis, but an appropriate level is required to not affect branching.

Given the high degree of gene conservation among cereals, the current state of knowledge facilitates a more detailed analysis of the development of reproductive structure in crops. Modern genome editing tools could be employed to target and manipulate CK levels to increase seed yield, with the concurrent aim of maintaining quality.

## 2. Future Directions

CKs are essential for reproductive success in plants. Currently, most knowledge regarding the role of CKs in reproduction involves the effect of mutation of genes involved either in CK metabolism or signalling (Figure 1, Table 1). To date, only a few non-ARR transcription factors, such as VDD, VAL, SEEDSTICK (STK), SPL and BEL1 [23,24,46] have been implicated to mediate CK responses downstream of CK signalling; therefore, the next step is to elucidate these downstream responses, to understand how CKs regulate specific developmental processes at the molecular level.

Another currently poorly studied aspect is how individual molecular CK species contribute to the biological effect of CKs. Some recent studies have highlighted the importance of specific CK species in eliciting specific biological responses [82], or have linked phenotypes with CK pools present in specific cellular compartments [23]. Due to the difficulty in accessing reproductive tissues, and particularly female reproductive tissues, studying these aspects should be coupled with microdissection and/or sorting techniques. Recently, some studies have quantified CKs in sorted cells [83]. This approach, coupled with more sensitive quantification techniques [84] and the development of biosensors [85], will undoubtedly advance our understanding of the cellular and subcellular distribution of CK species in reproductive tissues and their relationship to their developmental effects.

## Figures and Tables

**Figure 1 ijms-21-08161-f001:**
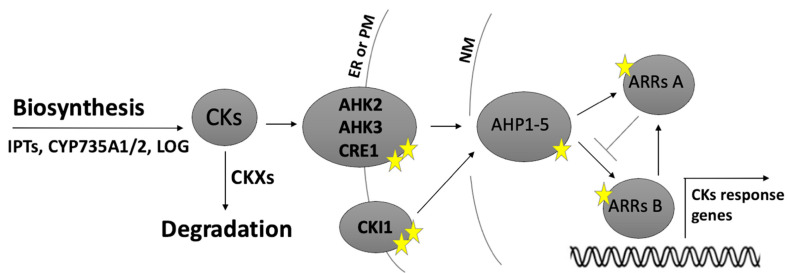
An overview of the cytokinin (CK) pathway from biosynthesis to response in Arabidopsis. CKs are synthesised by isopentenyltransferases (IPTs) and CYP735A1/2 enzymes and converted into active forms by LONELY GUY (LOG) enzymes, whereas cytokinin degradation occurs mainly through CKXs. CKs are perceived by the AHK2, AHK3 and AHK4/CRE1 receptors, which initiate a phosphorylation signalling cascade involving AHPs, which phosphorylate and activate type-B ARRs. Active ARRs then induce cytokinin-responsive genes, such as those encoding the type-A repressor ARRs. AHPs can also be phosphorylated by CKI, a histidine kinase that activates the cytokinin response in the absence of CKs. Stars represent phosphorylation sites. (ER) endoplasmic reticulum; (PM) plasma membrane; (NM) nuclear membrane.

**Figure 2 ijms-21-08161-f002:**
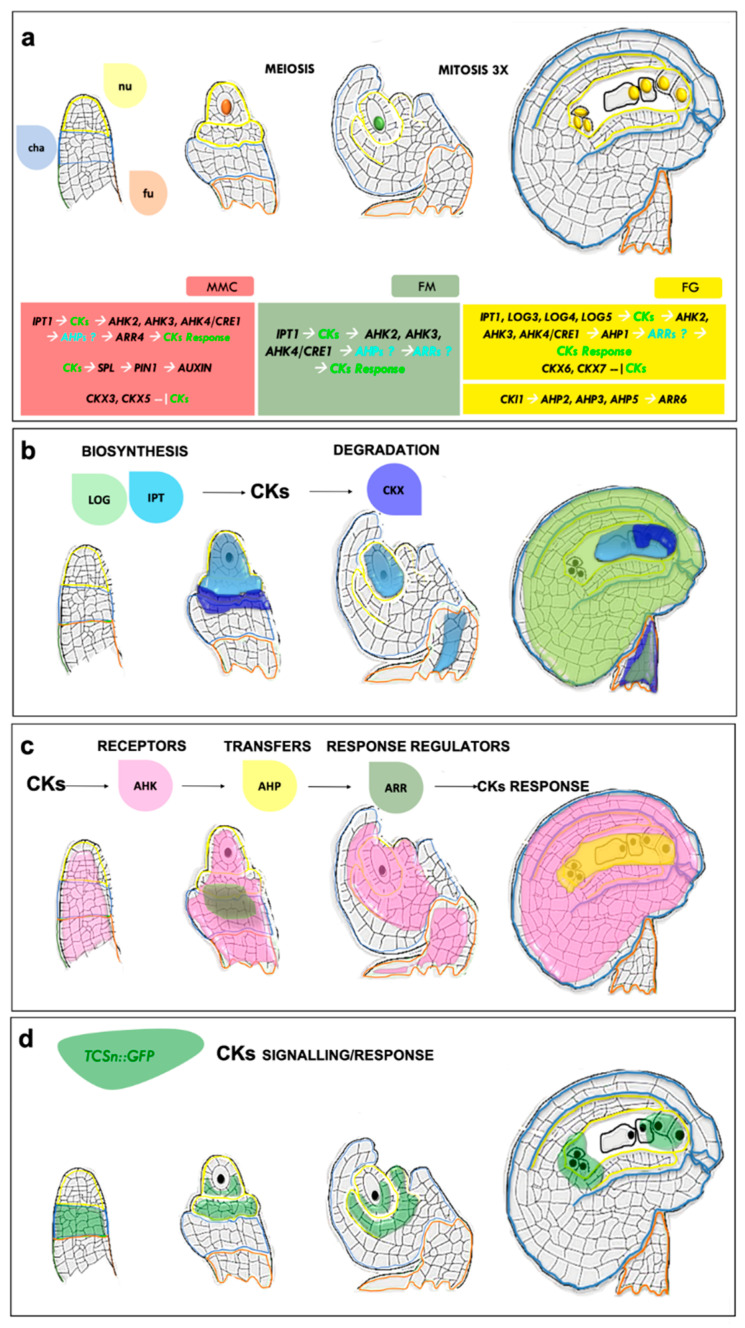
The expression domains of CKs-related genes during ovule development. CKs biosynthesis, degradation and signalling components expression domains and their hypothetical relationships. (**a**) Cartoon of the distinct and sequential phases of ovule development in Arabidopsis, first finger-like structure with three different zones outlined nucella, chalaza and funiculus, then ovule primordium with a megaspore mother cell (MMC) differentiated that enters in meiosis forming the functional megaspore (after the degeneration of the other 3 spores) that after three rounds of mitosis forms the mature female gametophyte (FG). Below, the distinct relationships among the different CK-related biosynthetic, degradation and signalling components are depicted in coloured rectangles: the MMC stage is highlighted in red, the FM stage is depicted in green and the mature female gametophyte stage in yellow. The hypothetic missing CK-elements are depicted with a ‘?’ and highlighted in light blue. (**b**) CK biosynthesis and degradation related genes expression domains: *pIPT1::GUS* is expressed at MMC and FM stages in the nucellus; at the mature stage of FG, it is active in the synergids and egg cells; the promoter of *LOG3* is active at FG stage in the funiculus, and *LOG4* at the same stage in the mature ovule integuments at the chalazal pole, whereas the activity *pLOG5::GUS* was detected at FG stage. Regarding CK degradation components at MMC stage the mRNA of *CKX5* was detected in the nucellus and chalaza. At FG stage *pCKX6::GUS* is restricted to the funiculus, whereas the *CKX7* promoter is active in synergid cells and CKX7 protein is detected in the synergids and egg cells. In light blue are IPT, in green the LOG and in dark blue the CKX expression domains. (**c**) The expression domains of CK signalling genes. The *AHK1* and *AHK3* promoters are active preferentially at the chalaza pole and only *AHK2* is expressed throughout the whole ovule development. Regarding ARRs activity, *ARR4* is present at MMC stage at the chalazal pole. In pink are depicted the AHK, in yellow the AHP and in green the ARR expression domains. (**d**) CK signalling/response is reported by *TCSn::GFP* expression: *GFP* expression is restricted to the chalazal region from during all stages of ovule development; at FG stage the signal is also present in the micropilar region. The region outlined in yellow is the nucellus (nu), blue represents the chalazal (chal) and the orange corresponds to the funiculus (fu). The ovule drawings were adapted from [24].

**Figure 3 ijms-21-08161-f003:**
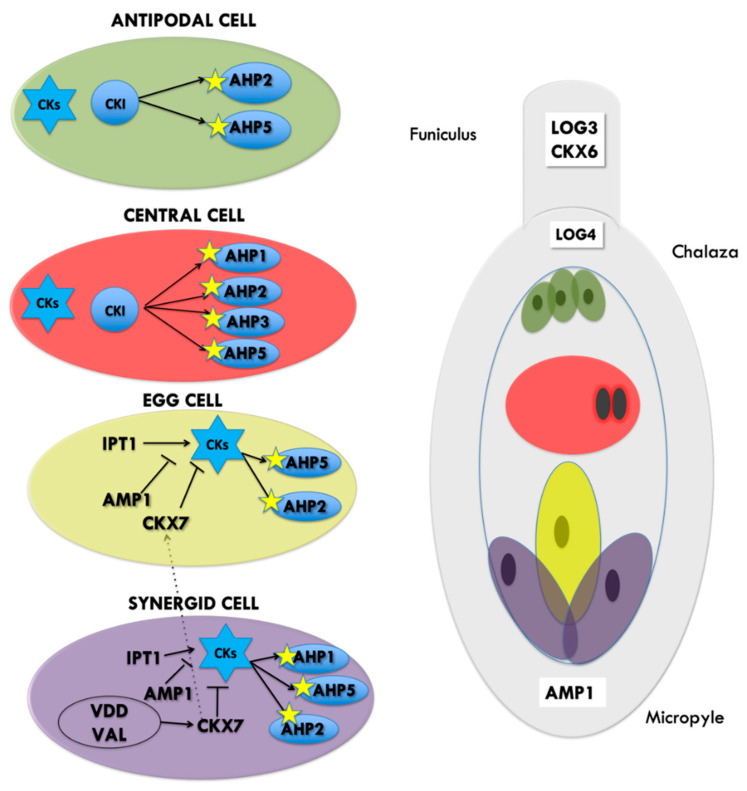
Arabidopsis mature female gametophyte and CKs involvement. The specific localisation and different relationships among CK biosynthesis, degradation and signalling components within the mature FG determine the identities and functions of each cell of the embryo sac. The signalling component *AHP1* is present in the central cell and the synergid cells; instead, *AHP2* and *AHP5* are expressed in all the gametophytic cells and *AHP3* is specifically localized in the central cell. CKI1, cytokinin-independent kinase is specifically expressed in the antipodals and central cells, interacting with the present AHPs. The components involved in CK biosynthesis and degradation genes, *IPT1* and *CKX7,* respectively, are mainly active on the opposite side of the embryo sac, towards the micropyle. Intriguingly, CKX7 promoter activity is detected in synergid cells and the protein in the egg cell. Components of CK biosynthesis and degradation are mainly expressed in the sporophytic tissues surrounding the mature ovule such as *LOG3* that is expressed in the funiculus, as well as degradation components, *CKX6.* Instead, *LOG4* that is expressed in the integuments at the chalaza side. *AMP1* is expressed in the ovule integuments in the synergids and weekly in the egg cell, AMP1 controls the content of CKs. *VDD* and *VAL* REM transcription factors are expressed both in synergid cells and control directly and/or indirectly the expression of *CKX7*. In purple are distinguished synergid cells, in yellow the egg cell, in red the central cell, in green the antipodals and in grey the sporophytic tissues surrounding the female gametophyte. Yellow stars represent the phosphoryl group. (CKs) cytokinins.

**Figure 4 ijms-21-08161-f004:**
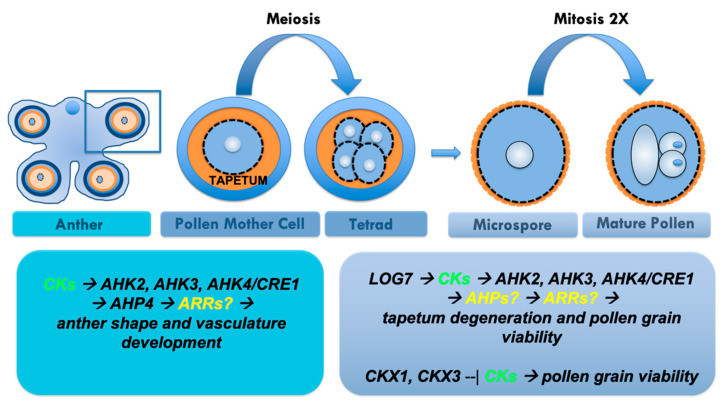
CK biosynthesis, degradation and signalling components during stamen and pollen development. This cartoon depicts the most important phases of pollen development inside Arabidopsis anther. The first step begins when the diploid pollen mother cell enter meiosis and give rise to a tetrad of haploid microspores that subsequently enter two rounds of mitosis: first round is asymmetrical forming the vegetative and germ cells, the second symmetrical only in the germ cell and gives rise to the sperm cells. The involvement of CKs in pollen development starts at the beginning of anther formation because anthers have to acquire a lobed shape and CK receptor mutants exhibit defects in anther shape. Subsequently, CK signalling components AHK2, AHK3, AHK4/CRE1 are also involved in regulating tapetum degeneration that influences ultimately the number of viable pollen grains. Likewise, CK degradation components CKX1 and CKX3 are involved in the viability of pollen grains.

**Table 1 ijms-21-08161-t001:** Cytokinin-related mutants with reproductive development defects in Arabidopsis.

Gene Name	Family or Protein Name	CK Process	Mutant Line	Phenotype	Ref
*LOG2*, *LOG3*, *LOG4, LOG5, LOG7, LOG8*	cytokinin riboside 5′-monophosphate phosphoribohydrolases	biosynthesis	*35S::LOG2,* *35S::LOG3,* *35S::LOG4,* *35S::LOG5,* *35S::LOG7,* *35S::LOG8*	Larger and heavier seeds	[53]
*CKX1*	CKX-cytokinin oxidase/dehydrogenase family	degradation	*35S:AtCKX1*	Few and enlarged seeds	[11]
*CKX3*	CKX-cytokinin oxidase/dehydrogenase family	degradation	*35S:AtCKX3*	Few and enlarged seeds	[11]
*CKX7*	CKX-cytokinin oxidase/dehydrogenase family	degradation	*ckx7* *ckx7 T*	Shorter fruit	[23]
*CKX3, CKX5*	CKX-cytokinin oxidase/dehydrogenase family	degradation	*ckx3 ckx5*	Increased production of flowers, longer siliques, more ovules and seeds	[20]
*CKX3, CKX5, AHP6*	CKX-cytokinin oxidase/dehydrogenase family AHP-Arabidopsis histidine phosphotransfer gene family	degradation signalling	*ckx3 ckx5 ahp6*	More siliques	[20]
*AHK4/CRE1, AHK2 and AHK3*	AHK-Arabidopsis histidine/kinase receptor family	signalling	*ahk2-1 ahk4-1* *ahk3-1 ahk4-1*	No anther dehiscence, small amount of viable pollen	[63]
			*cre1-12 ahk2-2tk ahk3-3*	Unable to produce seeds, abnormal development of female gametophyte and anthers; defects in tapetum degeneration and reduced number of pollen grains	[26]
			*cre1-12 ahk2-2 ahk3-3*	Abnormal female gametophyte development	[24]
			*ahk2-5 ahk3-7 cre1-2*	Reduced seed set, larger seeds	[69]
			*ahk2–7 ahk3 –3 cre1–12*	Defects in functional megaspore specification, female gametophyte absent	[25]
			*ahk2–1 ahk3–1 ahk4–1*	Sterile	[63]
*CKI1*	CKI-cytokinin-independent kinases	signalling	*cki1-5* *cki1-6*	Reduced seed set and abnormal embryo sacs	[36]
			*cki1-8*	Reduced seed set and abnormal embryo sacs	[35]
			*cki1-9*	Almost sterile, fewer larger seeds. Misspecification of female gametophyte cells	[37]
*AHP4*	AHP-Arabidopsis histidine phosphotransfer family	signalling	*35S::AHP4*	Anthers lack of secondary cell wall thickening in the anther endothecium	[64]
			*ahp4*	Anthers more lignified	[64]
*AHP1*,*2*,*3*,*4*,*5*	AHP-Arabidopsis Histidine phospotransfer family	signalling	*ahp 1, 2-1, 3/+, 4, 5-1*	Reduced seed set, but larger seeds; arrested during female gametophyte development	[41]
			*ahp1,2-2,3,4,5-1*	Loss of central cell fate and acquisition of egg-cell identity	[37]
*AHP2*,*3*,*5*	AHP-Arabidopsis Histidine phospotransfer family	signalling	*ahp2-2, ahp3, and ahp5-2*	Unfertilized ovule and seed abortion	[40]
*ARR1*,*2*,*10*,*12*	ARR-Arabidopsis response regulator type-B	signalling	*arr1–3 arr2–2 arr10–2 arr12–1*	Ovules arrested at FG7, the final developmental stage of the female gametophyte	[25]
*ARR1,10,12*	ARR-Arabidopsis response regulator type-B	signalling	*arr1-3 arr10-5 arr12-1*	Shorter siliques, larger seeds	[70]
*ARR7*,*15*	ARR-Arabidopsis response regulator type-A	signalling	*arr7arr15*	Female gametophyte lethal	[44]
*CRF5,6*	CRF-cytokinin response Factor	signalling	*crf5 crf6*	Homozygote non-viable	[17]
*CRF2,3,6*	CRF-cytokinin response Factor	signalling	*crf2 crf3 crf6*	Reduction in ovule number and placenta length	[29]
*AMP1*	Glutamate carboxypeptidase family		*amp1-1*	Increased cytokinin level	[49]
			*amp1-2*	Increased cytokinin level	[49]
			*amp1-10*	Supernumerary egg cell, twin embryos	[50]
			*amp1-13*	Supernumerary egg cell, twin embryos	[50]

**Table 2 ijms-21-08161-t002:** Cytokinin-related mutants with reproductive development defects in crop species.

Gene Name	Family or Protein Name	CK Process	Specie	Mutant Line	Phenotype	Ref
CKX1	CKX-cytokinin oxidase/dehydrogenase family	degradation	Zea Mays	pZmg13::CKX1 pZtap::CKX1	Defects in anther and stamen development	[75]
OsLOG-1	LOG- Lonely Guy	biosynthesis	*Oriza sativa*	Oslog-1	Flowers with only one stamen/no pistil	[77]
OsLOG-3	LOG- Lonely Guy	biosynthesis	*Oriza sativa*	Oslog-3	Flowers with six stamens and a slender pistil lacking an ovule or no pistil/absence of organ differentiation in the ovule founder region	[76]
OsLOG-4	LOG- Lonely Guy	biosynthesis	*Oriza sativa*	Oslog-4	Flowers with only one stamen/no pistil	[77]
OsCKX2	CKX-cytokinin oxidase/dehydrogenase family	degradation	*Oriza sativa*	Osckx2	Increased number of reproductive organs/enhanced grain yield	[80]
OsRR type-B	ARR-Arabidopsis response regulator type-B	signalling	*Oriza sativa*	rr24	Compromised anther development	[78]
OsRR type-B	ARR-Arabidopsis response regulator type-B	signalling	*Oriza sativa*	rr21/22/23	Compromised pollen capture	[78]
GSN1	GRAIN SIZE AND NUMBER		*Oriza sativa*	gsn1	Upregulation of cytokinin degradation enzyme, *CKX2*. Increase in grain size/less-branched inflorescence	[79]
TaCKX6-D1	CKX-cytokinin oxidase/dehydrogenase family	cytokinin degradation	Wheat (*Triticum aestivum*)	TaCKX6-D1 naturally occurring wheat variants		[81]

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
