# Peer review of "The Importance of Cytokinins during Reproductive Development in Arabidopsis and Beyond"

_ijms, 2020, doi:10.3390/ijms21218161_

Round 1

Reviewer 1 Report

The review by Terceros et al has summarized the roles of CKs in plant reproductive systems and pointed out some of the gap areas for further exploration. It reads well and would have broad range of readers in plant biology. Here are some comments/concerns.

It is understandable to use the model plant Arabidopsis for revisiting the roles of CKs, in this case, particularly reproductive development, I would suggest the authors to narrow a bit by changing the title to “The importance of cytokinins during reproductive development in Arabidopsis and beyond”. Though in the last section the authors have extended to crops such as Brassica, Maize and Rice, the roles of CKs in the regulation of reproductive development in other non-model or non-crop systems may be operating differently.

Are the figures in Figure 2 home made, they seem to lose focus, low resolution? if not home made, citations are in need.

line 108-110: check this sentence

line 258-259: Two Component Signaling Sensor (TCS)?

line 264: A detailed analysis of TCSn, TCS new?

line 269-271: check this sentence

line 301-302: check this sentence

line 302-304: check this sentence

line 366-367: check this sentence

line 387-390: check this sentence

line 433: type-B response regulators (RRs), were they orthologous to ARRs?

Author Response

The authors would like to thank the reviewer for the valuablecomments and suggestions to improve our manuscript. All suggestions made by the reviewer were gratefully accepted by the authors. The review by Terceros et al has summarized the roles of CKs in plant reproductive systems and pointed out some of the gap areas for further exploration. It reads well and would have broad range of readers in plant biology. Here are some comments/concerns.It is understandable to use the model plant Arabidopsis for revisiting the roles of CKs, in this case, particularly reproductive development, I would suggest the authors to narrow a bit by changing the title to “The importance of cytokinins during reproductive development in Arabidopsis and beyond”. Though in the last section the authors have extended to crops such as Brassica,Maize and Rice, the roles of CKs in the regulation of reproductive development in other non-model or non-crop systems may be operating differently.We thank the reviewer for the title suggestion, we have change accordingly.Are the figures in Figure 2 home made, they seem to lose focus, low resolution? if not home made, citations are in need. We thank the reviewer for the observation; indeed the ovule developmental drawings were adapted from a previous publication, the right citation was included in the figure legend.line 108-110: check this sentence “The genetic network that controls ovule development contains two other twokey transcription factors, BELL1 (BEL1) and SPOROCYTELESS/NOZZLE (SPL/NZZ), and the bel1 spl double mutant possesses finger-like ovule structures without integuments, similar to those in ahk2-2 ahk3-3 cre1-12 [24].”We have eliminated the repetition.line258-259: Two Component Signaling Sensor (TCS)?We thank the reviewer for the observation, the TCS isaCKs signallingmarker line already published and the meaning of TCS was already explained;we have eliminated the words “TwoComponent Signaling Sensor” and left only TCS with the correct referenceline 264: A detailed analysis of TCSn, TCS new? This refers to a NEW marker line for CK signallingthat was developed in 2013 by Zürcher and colleagues it is an improved version of the TCSsynthetic reporter for that TCS new (TCSn) again the explanation for this is in the original paper so we left only the abbreviation with the correct definition.Wehave changed “A detailed analysis...” with “The analysis ....”line 269-271: check this sentence “The same pattern of expression is observed during mitosis and gametogenesis; namely, GFP signal is detected in the sporophytic tissues in the chalazal region.At the mature ovule stage, when the FG is completely formed, GFP expression also becomes detecTable 1n central and antipodal cells.”We have corrected the misspelled words.

line 301-302: check this sentence “...cascade might be downregulated in amp1 ovules and prevent an increase the TCSn::GFP signal, even in the presence of higher levels of CKs. However, it cannot be excluded that AMP1 might contribute.”We have edited the sentence.line 302-304: check this sentence “...in the presence of higher levels of CKs. However, it cannot be excluded that AMP1 might contribute to synergid specification via a mechanism unrelated to CKs, as has been previouslydescribed for the SAM, were AMP1 regulates the stem-cell niche in a CK-independent manner (Huang et al., 2015).”We have edited the sentence.line 366-367: check this sentence CK gene network involved in these developmental processes largely remains unidentified. future studyshould include further understanding the crosstalk between CKs and other hormones, such as jasmonic acid and brassinosteroids, and the interactions among hormone signals and environmental cues.”We have edited the sentence.line 387-390: “....larger seeds than those in wild-type. When 35Spro:LOG4 plants were crossed with wild-type pollen, they produced seeds as large as those resulting from self-pollination, whereas seeds produced from the reciprocal wt mammae polline 35 cross were wild type, suggesting that CK activity in the sporophyte controls seed size [52].”We have edited the sentence.line 433: type-B response regulators (RRs), were they orthologous to ARRs?The answer is Yes, according to Yu-Chang Tsai and collaborators 2012 https://doi.org/10.1104/pp.111.192765.Although they discuss in the article that some OsRRs display a higher degree of lineage-specificity, and some can rescueArabidopsis phenotypes and others no.

Reviewer 2 Report

Terceros et al. present a review of the role of cytokinins in the development of male/female gametophytes and seeds, focusing on examples from Arabidopsis. They go through the effects of cytokinin on ovule development, cell identity in the female gametophyte, communication between sporophytic and gametophytic tissue, stamen development, pollen viability, seed production and, finally, describe some examples of the effects of cytokinins in crop plants.

This was an impressive and comprehensive compilation of primary literature on the role of cytokinins in plant reproductive development. On the other hand, a shortfall of this paper is that in certain parts of the paper, it goes into excruciating detail about every phenotype of every mutant study instead of distilling, organizing and interpreting the primary literature to weave a cohesive story for the reader. For example, there is an 80-line paragraph spanning lines 152 to 228. I think that the authors should go over this manuscript carefully to reduce the verbosity of the text. As it stands, I think this manuscript would be difficult to read for people that are not specialized in the role of cytokinins in plant reproductive development.

Here are some specific comments:

Figure 1: I think the stars represent phosphorylation but this was not explained in the legend.

Line 88: "FG" already defined on previous page.

Line 99: "CRE1/AH4" should be "CRE1/AHK4"

Line 270-271: "detectable" incorrectly written as "detecTable1n".

Figre 2: This was a difficult figure to understand though I think the content is useful
- Fig 2b. In the mature embryo sac, the central cell and egg cells look different from those in panels A and C. What is the reason for this? It seems to be an error unless I missed something.
- Lines 145-150 should be in Fig 2 legend.
- Consider making this a four panel figure. B-D would be the current A-C. The non-expression related annotations currently in panel C would be moved to the new panel A, which will not show any expression patterns, but only the developemental and tissue labels.

Line 337: should be "do not dehisce" as written in line 339 below it.

Table 1 and table 2:
- Make caption more desciptive. Something like "Cytokinin-related mutants in Arabidopsis defective in reproductive developemnt"
- In the column, "Family or protein name", text such as "cytokinin biosynthesis" or "cutoinin degradation" should be moved to a separate, new column.
- For Table 2 only, include a new column for species.

Author Response

We appreciate the reviewer’s positive and encouraging feedback. We have modified the manuscript to address the reviewer’s criticisms, andhope that these changes are sufficient and acceptable.Terceros et al. present a review of the role of cytokininsin the development of male/female gametophytes and seeds, focusing on examples from Arabidopsis. They go through the effects of cytokinin on ovule development, cell identity in the female gametophyte, communication between sporophytic and gametophytic tissue, stamen development, pollen viability, seed production and, finally, describe some examples of the effects of cytokinins in crop plants.This was an impressive and comprehensive compilation of primary literature on the role of cytokinins in plant reproductive development. On the other hand, a shortfall of this paper is that in certain parts of the paper, it goes into excruciating detail about every phenotype of every mutant study instead of distilling, organizing and interpreting the primary literature to weave a cohesive story for the reader. For example, there is an 80-line paragraph spanning lines 152 to 228. We do agree with the reviewer that this section is particularly specific, so we have eliminated few phrases and divided the big paragraph intosmaller ones, we hope that now this paragraph is less specific and more understandable.I think that the authors should go over this manuscript carefully to reduce the verbosity of the text. As it stands, I think this manuscript would be difficult to read for people that are not specialized in the role of cytokinins in plant reproductive development.We do agree with the reviewer that the review is very specific for CKs and reproduction in order to make it easier to read, each time that a gene family is mentioned we added the info in which CK process is mentioned Figure 1: I think the stars represent phosphorylation but this was not explained in the legend. We thank the reviewer for the observation, Stars are indeed phosphorylation sites we have now included in the legend.Line 88: "FG" already defined on previous page.Thank you, we deleted the definition.Line 99: "CRE1/AH4" should be "CRE1/AHK4"We have edited the text.Line 270-271: "detectable" incorrectly written as "detecTable1n".We have edited the text.Figure 2: This was a difficult figure to understand though I think the content is useful-Fig 2b. In the mature embryo sac, the central cell and egg cells look different from those in panels A and C. What is the reason for this? It seems to be an error unless I missed something.We thank the reviewerfor the observation, they should all be the same and so we have edited the figure accordingly.

Lines 145-150 should be in Fig 2 legend.We have edited the text and include it in the caption.-Consider making this a four panel figure. B-D would be the current A-C. The non-expression related annotations currently in panel C would be moved to the new panel A, which will not show any expression patterns, but only the developemental and tissue labels.We thank the reviewer for the suggestion the figure was updated accordingly.Line 337: should be "do not dehisce" as written in line 339 below it.We have edited the textas suggested.Table 1 and table 2:-Make caption more desciptive. Something like "Cytokinin-related mutants in Arabidopsis defective in reproductive developemnt"We thank the reviewer for the suggestion,we have improvedthe caption.-In the column, "Family or protein name", text such as "cytokinin biosynthesis" or "cutoinin degradation" should be moved to a separate, new column.We thank the reviewer for the suggestion, we have improved the table.-For Table 2 only, include a new column for species.We thank the reviewer for the suggestion, we have improved the table.

Reviewer 3 Report

This review focus on the current understanding of cytokinins signalling network during reproductive organs development. I found that the review was interested and the authors did a good job in explaining the complexcity of cytokinins signalling. Although it is a good review, I found the way the review structure was not easy to follow. 

There are many gene families that involves in cytokinins synthesis, feed back control, perceptions and transports. I would suggest that the authors perhaps add sub heading that highlight the roles of the gene families that involves in specific developmental process discuss in the review. This should help the readers especially the readers who just start working in the field that may not be familiar with the acronym of the gene families. 

Author Response

We appreciate the reviewer’s constructive and positive feedback. As suggested after mentioning each gene family we described in which part of CKprocessthey are involved in order to simplify the reading. We do agree that would be easier to give a special attention to the gene families that have huger impact in each of the reproductive phases described although as it is seen from the mutant analyses, there is more than one gene family involved in each phase and so it is very complex to choose which one is the “ representative”. We have also slightly change figure 2 that probably wasthe most complex.We hope that the changes that we did are beneficial for the reading, we believe that the article improved with the new changes.

Round 2

Reviewer 2 Report

Authors have addressed my comments from the first review.